Effect of modified pomace on copper migration via riverbank soil in southwest China

Chen Lingyuan 1
http://orcid.org/0000-0002-8792-1851 Abbas Touqeer 2
Yang Lin 1
Xu Yao 1
Deng Hongyan 1
Hou Lei 3
Li Wenbin 1 lwb062@163.com
1 College of Environmental Science and Engineering, China West Normal University , Nanchong , China
2 Zhejiang Provincial Key Laboratory of Agricultural Resources and Environment , Hangzhou , China
3 College of Resources & Environment, Tibet Agricultural and Animal Husbandry University , Nyingchi , China
Mortimer Monika
Electronic publication date: 2021 Jul 27
Publication date: 2021
Volume: 9
Electronic Location ID: e11844
Received 2020 Oct 8; Accepted 2021 Jul 1
Copyright: © 2021 Chen et al.
Copyright year: 2021
Copyright holder: Chen et al.
License: This is an open access article distributed under the terms of the Creative Commons Attribution License, which permits unrestricted use, distribution, reproduction and adaptation in any medium and for any purpose provided that it is properly attributed. For attribution, the original author(s), title, publication source (PeerJ) and either DOI or URL of the article must be cited.
License URL: https://creativecommons.org/licenses/by/4.0/

Keywords: Copper, Modified pomace, Riverbank soil, Geochemical characteristics, Column experiment

Funding: China West Normal University 17E062 Education Department of Sichuan Province 18ZB0576 Sichuan Province Science and Technology Support Program 2018JY0224 National Natural Science Foundation of P.R. China 41271244 This work was supported by the Fundamental Research Funds of China West Normal University (17E062), the Scientific Research Fundation of the Education Department of Sichuan Province (18ZB0576), the Sichuan Province Science and Technology Support Program (2018JY0224) and the National Natural Science Foundation of P.R. China (No. 41271244). The funders had no role in study design, data collection and analysis, decision to publish, or preparation of the manuscript.

==============================
To explore the effects of modified pomace on copper migration via the soil on the banks of the rivers in northern Sichuan and Chongqing, fruit pomace (P) and ethylene diamine tetra-acetic acid (EDTA) modified P (EP) were evenly added (1% mass ratio) to the soil samples of Guanyuan, Nanbu, Jialing, and Hechuan from the Jialing River; Mianyang and Suining from the Fu River; and Guangan and Dazhou from the Qu River. The geochemical characteristics and migration rules of copper in different amended soils were simulated by column experiment. Results showed that the permeation time of copper in each soil column was categorized as EP-amended > P-amended > original soil, and the permeation time of amended soil samples at different locations was Jialing > Suining > Mianyang > Guangan > Dazhou > Nanbu > Guanyuan > Hechuan. Meanwhile, the average flow rate of copper in each soil column showed a reverse trend with the permeation time. Copper in exchangeable, carbonate, and iron–manganese oxide forms decreased with the increase of vertical depth in the soil column, among which the most evident decreases appeared in the carbonate-bonding form. The copper accumulation in different locations presented a trend of Jialing > Suining > Mianyang > Guangan > Dazhou > Nanbu > Guangyuan > Hechuan, and the copper content under the same soil showed EP-amended > P-amended > original soil. The copper proportion of the carbonate form was the highest in each soil sample, followed by the exchangeable form. The proportions of iron-manganese oxide and organic matter forms were relatively small. A significant correlation was observed between the cation exchange capacity and the copper content in exchangeable and carbonate forms. Moreover, total organic carbon and copper contents were negatively correlated.

Introduction

With the continuous innovation and development of modern industries, human society has made progress. However, the pollution threat to the environment has become increasingly serious (Pérez et al., 2006; Tahmineh et al., 2013). A large number of harmful metals (i.e., Hg, Cd, Pb, Cu, Cr, and Zn) are discharged into the soil environment (Huang et al., 2007; Bhuiyan et al., 2010), thus damaging the soil microbial community, altering the soil chemical properties, incorporating into plants, and ultimately entering the food cycle. Among these metals, the toxicity of copper pollution cannot be underestimated. Copper pollution mainly comes from natural and anthropogenic activities, and it cannot be degraded in the soil. Compared with other heavy metal elements, copper pollution is more common (Xiong et al., 2010). It not only brings great difficulties to soil remediation but also contaminates bodies of water (Fereydoun et al., 2015), which directly or indirectly poses threats to biological and human beings. In recent years, heavy metal pollution has become increasingly serious (Zhu et al., 2018; Reetu et al., 2019). Similarly, the application of livestock and poultry manure on the banks of the Jialing River, the Fu River, and the Qu River in northern Sichuan and Chongqing has caused severe copper pollution in the soil (Mohammad, 2017; Li et al., 2012). Riverbank soil is the last barrier of the river from copper pollution. Therefore, studying this soil is of great significance to determine the migration features and migration rules of copper in the bank soil to protect the river body.

Studies on the application of materials for repairing copper-contaminated soil have been widely praised (Bes & Mench, 2008; Malandrino et al., 2011). Mao et al. (2019) found that various factors, such as ash content, fixed carbon content, specific surface area, oxygen–carbon ratio, and pore size distribution, affect the copper adsorption capacity by wood biochar (Wang et al., 2019). Shen, Lei & Chen (2017a) used NaOH/KOH modified biochar (prepared by lemon residue) to adsorb copper ion and found that the adsorption rate of copper was 7.28–8.40%. Thus, the influence of NaOH/KOH modification on copper adsorption was not remarkable (the adsorption rate was 5.04% of unmodified biochar). Other researchers used H3BO3 (Shen, Lei & Shao, 2017b) and H3PO4 (Shen & Lei, 2016) modified lemon residue to carry out the adsorption experiment, and the copper adsorption rate reached 29.71% and 26.90%, respectively. Lemon pomace contains citric acid, which can neutralize NaOH/KOH, and enhances the adsorption ability of biochar. Moreover, acid-modified material can better improve the adsorption ability to copper than alkaline-modified material. Ethylene diamine tetra-acetic acid (EDTA) is an organo-chemical agent and chelating agent. Previous studies have shown that EDTA has a wide range of coordination properties and can form stable chelates with almost all metal ions (Yoshinobu et al., 1997; Radanovi et al., 2004). The removal effect of EDTA on soil heavy metals is remarkably higher than that of other surfactants (Wasay, Barrington & Tokunaga, 1998; Zhang et al., 2013). EDTA can form a EDTA-Cu chelate with high stability (Yuan et al., 2019), which is a very favorable modifier for lemon residue modification used in copper adsorption.

Anyue City of Sichuan province enjoys the reputation of being “the hometown of Chinese lemon” and stands out in the development of the Chinese lemon industry. The lemon planting area of Anyue City was 34,700 hm2 in 2018. In addition, the fresh fruit output was 580,000 T, which accounted for more than 80% of the total area and output of the city (Li & Zhu, 2019). The products developed from lemon production include lemon oil, lemon fermented vinegar, lemon pectin, lemon beverage, lemon fermented fruit wine, and lemon tea. A large amount of waste lemon pomace residue is produced in deep processing every year (Qin, Zhou & Cao, 2012; Shen & Shen, 2012). The traditional treatment method of lemon pomace involves directly discarding, landfilling, or processing the product into feed, which not only wastes resources but also has a negative impact on the environment (Zhang et al., 2006; Li et al., 2011; Hao et al., 2016; Li & Zhu, 2019). Lemon pomace residue contains numerous functional groups that can chelate with metal ions or molecules, such as the carboxyl group, hydroxyl group, thiol group, and amino group (Parajuli et al., 2008). Using lemon pomace as an adsorbent not only realizes the comprehensive utilization of lemon pomace but also absorbs the heavy metal in the soil to achieve the purpose of soil restoration.

As the transition zone between land and river, riverbank soil plays an important role in the adsorption and retardation of pollutant migration into river water. With the long-term discharge of industrial and agricultural production wastes on the banks of the rivers, heavy metal pollutants in riverbank soil are accumulating, which poses a serious threat to the river water (Li et al., 2020). In this study, lemon pomace and EDTA-modified lemon pomace were used to amend a riverbank soil sample. Then, the raw soil was taken as the control. The soil column model (Christian et al., 1999; Papassiopi, Tambouris & Kontopoulos, 1999; Hu et al., 2014; Sun et al., 2001) was selected to study the distribution, accumulation, migration, and geochemical characteristics of copper in the riverbank soil. The purpose of this study was to provide reference for the protection of river basins in northern Sichuan and Chongqing and to explore the benefits of fruit residual waste for the remediation of heavy-metal-contaminated soil.

Materials and Methods

Materials

Lemon pomace (P) was collected in Anyue City, Sichuan Province, and the samping work was conducted in privately-owned land from Mr. Xie Jia. The raw P was washed with deionized water several times, dried in an oven at 60 °C to constant weight, crushed, and sieved through 40 mesh size. Then, it was accurately weighed to 50 g into a 1-L beaker. Afterward 250 mL of absolute ethyl alcohol and 250 mL of 0.4 mol/L NaOH solution were added. Then, the solution was stirred for 24 h and washed again with deionized water to neutral, centrifugal filtration. Finally, it was put it into the oven for drying, and grounded through a 60-mesh sieve. After the materials were prepared, they were set aside for later use.

EDTA was used as the modifier to modify the lemon pomace. A Cu2+ solution was prepared by using analytical reagent-grade CuSO4·5H2O. EDTA (analytical reagent) and CuSO4·5H2O (analytical reagent) were purchased from Chengdu Kelon Chemical Reagent Factory, Chengdu City, Sichuan Province, China.

The specific process of EDTA-modified P (EP) preparation was as follows. First, 50 g of pretreated P material was accurately weighed into a 1-L beaker. Then, a 500-mL 0.5-mol/L EDTA solution was added. Next, the material was stirred evenly and put it in a constant temperature oscillator at 25 °C for 12 h. Then, it was washed to neutral with deionized water and centrifugally filtered. Finally, it was left in the oven for drying and grounding through a 60-mesh sieve.

The soil samples were collected from three rivers in Sichuan and Chongqing. The sampling points (Fig. 1) from north to south were Guangyuan (GY), Nanbu (NB), Jialing (JL), and Hechuan (HC) from the Jialing River; Mianyang (MY) and Suining (SN) from the Fu River; and Guang’an (GA) and Dazhou (DZ) from the Qu River. Down by the village and within 50 m from the riverbank, a typical area (with same vegetation type and land use pattern) was selected, and 0–20 cm soil samples were collected via multi-point sampling. The soil samples were mixed evenly, air-dried, and then passed through a 60-mesh sieve. The physical-chemical properties of the samples are shown in Table 1.

Figure 1 Distribution map of sampling points.

The sampling points are Guangyuan (GY), Nanbu (NB), Jialing (JL), and Hechuan (HC) from the Jialing River; Mianyang (MY) and Suining (SN) from the Fu River; and Guang’an (GA) and Dazhou (DZ) from the Qu River.

Table 1 Basic physical-chemical properties of the riverbank soils, the sampling points, GY, NB, JL, HC, MY, SN, GA and DZ are short for Guanyuan, Nanbu, Jialing, Hechuan, Mianyang, Suining, Guangan and Dazhou, respectively. The same as in other tables and figures.

The average values are given. Soil humidity in the table is divided into four grades: wet (water content ≥ 40%), moist (40% > water content ≥10%), slightly moist (10% > water content ≥ 5%), and dry (water content < 5%). The number of soil pores is divided into three grades: much (porosity ≥ 60%), medium (60% > porosity ≥ 30%), and little (porosity < 30%). The number of soil roots is divided into three grades: much (root denesty ≥ 20%), medium (20% > root denesty ≥ 5%), and little (root porosity <5%). Soil structure is divided into block, granular, flake, nucleus, and so on. The pH value was measured by HQ411D desktop pH meter of American Hash Company, the CEC (cation exchange capacity) was measured by sodium acetate ammonium acetate method, the SSA (specific surface area) was measured by V-Sorb2800P Specific surface area analyzer and the TOC (total organic carbon) was measured by TOC-VCPH TOC instrument.

Soil samples	Longitude and latitude	Altitude (m)	Crop rotation	Soil
humidity	Soil porosity	Soil
structure	Root number	pH	CEC
(mmol/kg)	TOC
(g/kg)	SSA
(m2/g)	Cu content
(mg/kg)	Clay content (%)	
GY	E105°55′18.94″
N31°45′6.12″	430.3	Peanu, corn	slightly moist	much	granular	little	8.16	120.72	15.75	90.34	18.84	10.74	
NB	E106°09′1.38″
N31°17′48.79″	427.4	Chili, rice	slightly moist	little	blocky	little	7.99	118.22	28.16	110.44	13.80	9.22	
JL	E106°06′37″
N30°42′19″	270.0	Corn, sweet potato	moist	much	granular	medium	7.70	204.08	12.28	130.21	16.32	12.51	
HC	E106°17′30″
N29°58′23″	245.1	Sweet potato, green beans	slightly moist	little	blocky	little	6.56	100.69	25.83	89.34	24.74	7.28	
MY	E105°03′25.6″
N31°08′57.43″	380.7	Corn, konjac	moist	little	blocky	little	7.80	122.53	25.01	98.25	19.68	9.27	
SN	E105°38′17″
N30°27′45″	283.1	Corn, pterocarpus	moist	little	blocky	much	7.44	182.79	8.84	103.14	8.75	10.66	
DZ	E106°57′20.07″
N30°51′38.5″	314.0	Corn, rice	slightly moist	little	granular	little	7.42	118.31	24.23	141.27	23.04	9.41	
GA	E106°40′25″
N30°27′42″	0.0	sweet potato, corn	slightly moist	medium	granular	much	7.59	153.98	21.56	102.45	58.38	10.15	

Experimental device

Riverbank soil has low clay content and higher sand content with a large particle diameter. In addition, Cu2+ migrates faster in riverbank soil than that in agriculture soil. By experimenting with the blocking effects and lateral diffusion of Cu2+ in soil columns of various sizes, a PVC pipe (length 20 cm, inner diameter 1.6 cm, and outer diameter 2 cm) was used to make a soil column for simulating the migration and distribution characteristics of copper. Sieve plate was placed at the inlet end and outlet end of the column (upper sieve plate prevents the water from impacting the soil sample, and the lower sieve plate prevents the soil sample from flowing down). Sand (0.3 cm) was filled between the upper sieve plate and the soil sample, and between the lower sieve plate and the soil sample to prevent the soil sample from blocking the micro-pores on the sieve plate. The flow rate of the Cu2+ solution was controlled by the flow meter in the pipeline. The specific schematic diagram is shown in Fig. 2.

Figure 2 Scheme of the experimental setup.

Experimental design

P and EP were added to the bank soil samples of GY, NB, JL, HC, MY, SN, GA, and DZ, at a mass ratio of 1%. They were mixed evenly with original bank soils as contrast (CK), thus resulting in 24 kinds of amended soil samples, namely, GYCK (GY soil), GYP (GY soil + 1%P), GYEP (GY soil + 1%EP), and so on. The column experiment was conducted to simulate the migration and geochemical characteristics of copper in each amended soil sample. Each treatment was repeated three times.

Experimental methods

The amended soil samples were uniformly loaded into the soil column to maintain the same weight and density, with a weight of 48 g, a density of 1.19 g/cm3, and water content of 1.5–2.2%. A fluid reservoir containing 500 mg/L of CuSO4·5H2O was placed 1 m above the soil column and connected by a hose. The Cu2+ solution switch was turned on to control the natural flow rate. The Cu2+ concentration at the outlet of the soil column was measured every day at same time until the Cu2+ concentration was equal to 500 mg/L. Then, the total leaching time and volume of the Cu2+ solution were determined. Next, the Cu2+ solution in the reservoir was replaced with deionized water to wash (12 h) the residual Cu2+ solution in the soil sample pores. After the experiment, the soil column was divided into four parts: 0–5 cm, 5–10 cm, 10–15 cm, and 15–20 cm from top to bottom. The soil samples of each part were air-dried and passed through a 60-mesh sieve. Then, the copper content in exchangeable, carbonate, iron–manganese oxide, and the organic matter forms of the four parts were determined (Ari & Anne, 2005; Silveira et al., 2006; Perlatti et al., 2014).

Cu2+ contents in soil samples were determined via Hitachi Z-5000 (Japan) flame atomic absorption spectrophotometry, and background absorption was corrected through Zeeman effect. The above determination was inserted into the standard solution for analysis and quality control.

SPSS 16.0 statistical analysis software was used to process the experimental data for correlation analysis. AutoCAD2020 and SigmaPlot 10.0 software was adopted to improve data plotting.

Results

Copper penetration characteristics

The results in Table 2 showed that as Cu2+ passed through the soil column, the permeation time of copper (when the Cu2+ concentrations at the inlet and outlet are identical) in each soil column was between 68 and 208 h. For the same soil samples, the time required for copper penetration equilibrium in the soil column showed the trend of EP-amended > P-amended > original soil (CK). In addition, the permeation time of EP- and P-amended soil is greater than that of CK. The average flow rate was negatively correlated with the permeation time, which showed a trend of CK > P-amended > EP-amended soil, and the average flow rate was between 9.7 and 18.9 mL/h. In general, the addition of P and EP in the soil prolonged the penetration time and reduced the penetration velocity. However, the comparative effect of the two materials showed that the adsorption performance of EP for copper ions was stronger than that of P. Given that EDTA is an acidic organic compound, it can chelate with almost all heavy metal ions to form a stable compound. Among the metal chelating agents, the chelating effect of copper is significant (Zhang et al., 2018).

Table 2 Permeation time and average velocity of Cu2+ in different soil columns.

Soil samples	Permeation time (h)	Average velocity (mL/h)	Soil samples	Permeation time (h)	Average velocity (mL/h)	
GY	GYCK	80	18.0	MY	MYCK	128	14.1	
GYP	122	15.3	MYP	156	12.8	
GYEP	132	13.0	MYEP	180	11.0	
NB	NBCK	92	17.4	SN	SNCK	140	13.4	
NBP	120	14.1	SNP	172	12.0	
NBEP	144	12.5	SNEP	192	10.3	
JL	JLCK	164	12.2	DZ	DZCK	108	16.4	
JLP	184	11.4	DZP	136	13.7	
JLEP	208	9.7	DZEP	152	11.7	
HC	HCCK	68	18.9	GA	GACK	116	15.6	
HCP	96	16.3	GAP	148	13.2	
HCEP	122	14.3	GAEP	164	11.1	
Note:

Pomace (P) and ethylene diamine tetra-acetic acid modified P (EP) were added to GY, NB, JL, HC, MY, SN, GA, and DZ with original bank soils as contrast (CK), namely, GYCK (GY soil), GYP (GY soil + 1%P), GYEP (GY soil + 1%EP), and so on.

For the same soil sample, the permeation time showed a higher to lower trend, i.e., JL > SN > MY > GA > DZ > NB > GY > HC, while the average velocity was negatively correlated with the permeation time (Fig. 3). The permeation time of the HC soil sample to reach equilibrium was the shortest, and the average flow rate was the largest. This outcome is mainly because the pH value of the HC soil sample was acidic; the free copper in the soil was increased, and the cation exchange capacity (CEC) of HC soil was small (Arias et al., 2006), which was not conducive to the adsorption and immobilization of copper in the soil column. However, the CEC and clay content of JL soil sample was higher (Table 1). Thus, the adsorption of copper was stronger, which was conducive to the adsorption and immobilization of copper.

Figure 3 Copper penetration characteristics in the soil columns.

CK, P, and EP stand for original, pomace-amended and ethylene diamine tetra-acetic acid modified P-amended soil. The same as other tables and figures. (A–H) The riverbank soils from GY, NB, JL, HC, MY, SN, GA and DZ, respectively.

Vertical distribution of copper

After the penetration test, the soil columns were divided into four parts: 0–5, 5–10, 10–15, and 15–20 cm from top to bottom. Copper in exchangeable, carbonate, and iron–manganese oxide form decreased from top to bottom, as shown in Fig. 4. Among all forms of copper, the carbonate form decreased most, followed by the exchangeable and iron–manganese oxide forms. The organic matter form was the weakest (Fig. 4). The distribution curve of copper on P-amended soil was relatively concentrated, while most of the copper distribution curve was dispersed on EP-amended soil. Copper in exchangeable and carbonate forms on CK changed remarkably compared with P- and EP-amended soil, which proved that P and EP played a certain role in altering the distribution rule of copper forms in the soil column.

Figure 4 Distribution characteristics of Cu in different forms in vertical section.

(A–C), (D–F), (G–I), and (J–L) were exchangeable, carbonate, iron–manganese oxide, and organic matter form, respectively.

Copper accumulation characteristics

After leaching, the cumulative copper content in the mixed soil samples ranged from 21.70 to 45.88 g/kg (Fig. 5). Furthermore, the copper content showed a trend of JL > SN > MY > GA > DZ > NB > GY > HC. The results were consistent with the trend of permeation time, which were contrary to the results of average velocity. Thus, a longer permeation time resulted in stronger copper adsorption and higher copper accumulation content in the soil column. The copper content in different soil samples showed a trend of EP-amended > P-amended > CK. The results demonstrated that the addition of P could enhance the ability of soil to accumulate copper. The main reason is that this addition increases the content of organic matter in the soil, thus enhancing the complexation of copper ions in the soil samples. Compared with P, EP-amended soil has stronger adsorption capacity for copper. This characteristic may be due to the increase in porosity and specific surface area of soil by the added material of P and EP, which provides a powerful condition for the adsorption of copper ions (Zhang & Zhan, 2012; Hu, An & Xu, 2016); In addition, EP can form metal complexes with heavy metals, which ultimately enhances the adsorption capacity of copper. The proportions of different copper forms in the soil samples varied. Among them, the carbonate form accounted for the highest at 46.01%, followed by the exchangeable form accounting for 25.96% and then the iron-manganese oxide form (16.52%). Finally, the organic matter form (11.51%) was relatively small. This proportion was mainly related to a higher amount of carbonates in the soil sample and the ability to exchange ions.

Figure 5 Copper contents (colour changes) in various forms of different riverbank soils from the three rivers.

(A–D), (E–H), and (I–L) indicate original, P-amended, and EP-amended soil, respectively.

Discussions

Correlation between soil physicochemical properties and Cu forms

Soil pH, CEC, total organic carbon (TOC), and specific surface area have important effects on the chemical forms of copper, as shown in Fig. 6. On the one hand, the results showed that the CEC of soil was significantly correlated with the contents of exchangeable copper and carbonate copper in different soils. Moreover, the clay content had a significant correlation with exchangeable copper. On the other hand, the correlation between pH and specific surface area and various forms of copper content in soil was non-significant. The results showed that CEC had a great influence on the copper content in soil samples, and pH and specific surface area had little impact on the copper in the samples. The above results were mainly because of CEC, which could provide the number of adsorption sites on the soil surface. The larger number of adsorption sites, the stronger the ions in the exchangeable form were (Lu, Wang & Ma, 2014). TOC and copper contents in the soil sample were negatively correlated. The TOC content in the soil sample occupied the adsorption point of the tested soil, thus inhibiting the adsorption capacity of the tested soil to copper (Teng et al., 2007)

Figure 6 Correlation analysis of soil physicochemical properties and Cu geochemical characteristics.

EX, CA, IMO, OM, and TO are the acronyms for exchangeable, carbonate, iron-manganese oxide, organic matter and total forms of copper, respectively. (A), (B), and (C) were original, P-amended, and EP-amended soil respectively. * and ** indicate a significant correlation at the level of p = 0.05 or 0.01. (r = 0.959 and r = 0.878 when the degree of freedom is f = 7).

Difference in copper adsorption

Our results showed the variations of copper adsorption on different tested soils (Fig. 7). The absorption of copper by natural soil depended on the physical and chemical properties of soil and clay content. The adsorption capacity of clay for copper mainly depended on the negative charge adsorption point between layers (Cui et al., 2010). However, the adsorption capacity of the original soil for copper was low because of their low clay content. P was mainly composed of cellulose, hemicellulose, and pectin. Research demonstrated that the metal adsorption affinity of P mainly came from the surface bonding of cellulose and complexing action of pectin (Mykola, Lidiya & Batyr, 1999). Therefore, P-amended soil in this study had higher copper binding capacity than the original soil. After modification by EDTA, the effective functional groups on P were increased, and the morphology was more porous (Qi, 2012). Meanwhile, EDTA was able to form stable complexes with most heavy metal ions (Zhang et al., 2018; Li, 2014), and improve the adsorption capacity of EP-amended soil samples for copper.

Figure 7 Adsorption differences of copper on different riverbank soils (original, P-amended, and EP-amended soil).

CK is original soil, P and EP were pomace and ethylene diamine tetra-acetic acid modified pomace.

Conclusion

This study showed several key points, which described the effectiveness of amendments on the immobilization of heavy metals. The permeation time of copper in each soil column showed EP-amended > P-amended > original soil. In addition, it presented JL > SN > MY > GA > DZ > NB > GY > HC at different locations. The average velocity of copper on each soil column indicated a reverse trend with the permeation time. In the vertical distribution, copper contents in exchangeable, carbonate, and iron–manganese oxide forms decreased with the increase of soil column depth, among which the carbonate form decreased most evidently. Copper accumulation at various locations and on different soils presented the same trend with permeation time. The largest proportion was the carbonate form, followed by the exchangeable and iron–manganese oxide forms. The organic matter form was the lowest. A significant correlation was observed between the CEC and the exchangeable and carbonate forms of copper. Meanwhile, TOC and copper contents were negatively correlated.

Supplemental Information

Supplemental Information 1 Data of Accumulation characteristics.

Click here for additional data file.

Supplemental Information 2 Data of Correlation.

Click here for additional data file.

Supplemental Information 3 Data of Penetration characteristics.

Click here for additional data file.

Supplemental Information 4 Data of Vertical distribution.

Click here for additional data file.

Additional Information and Declarations

Competing Interests

Author Contributions

Field Study Permissions

Data Availability

The authors declare that they have no competing interests.

Lingyuan Chen conceived and designed the experiments, performed the experiments, prepared figures and/or tables, authored or reviewed drafts of the paper, and approved the final draft.

Touqeer Abbas conceived and designed the experiments, analyzed the data, authored or reviewed drafts of the paper, and approved the final draft.

Lin Yang performed the experiments, authored or reviewed drafts of the paper, and approved the final draft.

Yao Xu performed the experiments, authored or reviewed drafts of the paper, and approved the final draft.

Hongyan Deng performed the experiments, analyzed the data, prepared figures and/or tables, and approved the final draft.

Lei Hou conceived and designed the experiments, performed the experiments, analyzed the data, authored or reviewed drafts of the paper, and approved the final draft.

Wenbin Li conceived and designed the experiments, performed the experiments, analyzed the data, prepared figures and/or tables, authored or reviewed drafts of the paper, and approved the final draft.

The following information was supplied relating to field study approvals (i.e., approving body and any reference numbers):

Lemon pomace (P) was collected in Anyue City, Sichuan Province, and the sampling work was conducted in privately-owned land from Mr. Xie Jia.

The following information was supplied regarding data availability:

The raw measurements are available in the Supplemental Files.

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
