# Peer review of "Effect of modified pomace on copper migration via riverbank soil in southwest China"

_PeerJ, doi:10.7717/peerj.11844_

## Round 0.1 · original submission · Major Revisions

I kindly ask you to address possible limitations of using a PVC column with a diameter of 1.6 cm and a length of 20 cm to simulate soil conditions. In other words, you should justify to the readers why you used this method.

Reviewer 1 ·

Basic reporting

The paper proposed to evaluate the use of fruit pomace, naturally and modified with EDTA, in the immobilization of Copper in areas close to rivers in different places in China. It is a valid assessment, in the sense of looking for “ecological” alternatives to minimize the impacts caused by human activities, with the use of residues for the treatment of contaminants and meets the premises of a circular and sustainable economy.
Initially I indicate that I identified several flaws in the writing of the article regarding the English language. Several sentences and words throughout the text are improperly written, which means that the article must undergo a complete revision of English (i.e. L47-48, 50, 60, 62-63, 139 ... and so on ).
The purpose of the work is clear, to test the different compounds of the pomace to assess their ability to immobilize or decrease the flow of copper from the soil to the rivers. However, several flaws are identified in the way that the results were presented and analyzed.

Experimental design

The experimental design used a column experiment, which is a methodology already well used to evaluate experiments of this type. However, some procedures and definitions can be improved.

First, why was lemon pomace used? Is it a common waste at the site? Easy to obtain? Or because other studies have already demonstrated the potential of this material? It is important to write down why this material was used specifically.

Between lines 92-94, is it described the characteristics of what reagents was used? I could not understand this paragraph. I believe it is only a description of the reagents used. It is? If so, write that paragraph better.

L.102 “Within 50 m from the riverbank, a typical area was selected”. What would these typical areas be? What are the human activities at these collection points? Was there the same activity at all points? Agriculture? Industry? Residence? I saw that table 2 indicates the cultures? You should mention this at this point.

L115-116. You don't need to repeat all locations again. Suffice to say that it was mixed with all the samples.

L120. “The column experiment was used to simulate the migration and morphological characteristics of copper”. The authors write "copper morphological characteristics". I think the most appropriate term would be geochemical characteristics.

L 130-131. “Then the exchangeable copper, carbonate, iron_manganese oxide, and organic matter of soil parts were determined.” After all, were four or five fractions analyzed? In addition, the method proposed by Tessier is an already established method of sequential extraction, but there are many other more modern methodologies in the literature that could have been considered.(i.e. (Gimeno-García et al., 1995;
Siregar et al., 2005; Silveira et al., 2006, Perlatti et al., 2014).

Validity of the findings

L. 140-141. The authors use the term "lemon dregs". Standardize the identification in the text. Pomace or dregs?

L.142-144. “The adsorption performance of lemon dregs modified by EDTA was stronger for copper ions, because of the acidic nature of lemon dregs after modification will enhance its adsorption capacity for copper ions Shen et al. (Shen and Lei, 2016; Shen et al., 2017b) ”. This statement is not based on the literature, since it is known that the greater the acidity, the greater the mobility of copper in the soil. Perhaps should be better to emphasize the chelating capacity of EDTA rather to address the greater acidity.

L 147-150. Very speculative stretch. It is very difficult to evaluate this statement without having provided the complete physical characteristics of the soil, such as clay, silt and sand content. These elements have the greatest influence on the dynamics of the soil solution and adsorption processes.

L172-173. “The results showed that the addition of lemon pomace could enhance the ability of soil to accumulate copper”. This seems a little obvious, given the well-known affinity of copper with organic matter. The authors could better explore the correlation between the increase in TOC levels with the addition of lemon pomace to identify if there was an increase in carbon levels, as it is a component with high affinity with Cu.

L 176-178. “EDTA modified fruit residue can form soluble metal complexes with heavy metals, which improve the migration rate of heavy metal ions in soil (Dariush et al., 2017), and ultimately enhances the adsorption capacity of copper”. This statement presents antagonistic data. If EDTA favors metal solubilization, how can it favor adsorption? The right thing would be to increase mobility and consequent leaching. I suggest that the authors explain this phenomenon better.

L187-188. Considering that the pH has a lot of influence on the mobility of copper, the authors should better explore this statement. Perhaps the fact that all soils are slightly alkaline may have aggravated this process.

L 188 - 191. This passage is confusing and difficult to understand. What would be a moderate correlation between Cu and carbon?

In general, I consider the section 2.4 very confusing. I suggest rewriting, pointing out specifically the role of each property (TOC, Surface area, carbonates, oxides) in Cu geochemical behaviour.

L 198-199 - “The absorption of copper by natural soil depends on the physical and chemical properties of soil and clay content.” I agree with that statement. But where are the data on soil clay content? And wouldn't it be adsorption instead of absorption?

L 201-205. This statement is very speculative. It must be associated if the presence of lemon pomace increases the contents of TOC in the soil, which in turn can help in the adsorption and immobilization of the metal.

Conclusions: The conclusion is not supported by the results presented. It is clear that CEC has a great influence on the adsorption of ions in the soil, since it is the general capacity of the soil to retain not only contaminants, but also macro and micro-nutrients. The authors must better correlate the natural components of the soil, the changes provided by the addition of the materials and the specific effects of these on the geochemical processes that influence the mobility of copper.

Additional comments

I believe that the paper has the potential to be published, as it presents interesting data and addresses a relevant topic.
However, the analysis and discussion of the data must be improved, with a more robust scientific basis. There are many studies that demonstrate the effects of pH and organic matter content on the geochemical dynamics of copper, and I believe that the authors should focus more on that.

With a little more depth discussion and better organization of topics, I believe it can be published.

Reviewer 2 ·

Basic reporting

The article should include a sufficient background to demonstrate why author choose modified materials for solidification of copper in a riverbank soil. I suggest that you improve the description at lines 63- 68 to provide research significance for your study (specifically, you should expand upon the practical value or research significance of your work). In addition, the English language expression should be improved to ensure that an international audience can clearly understand your text.

Experimental design

The manuscript "Effect of modified pomace on copper migration via riverbank soil in southwest China" is a study about leaching behaviour and remediation of heavy metal in riverbank soil in southwest China. The authors attempted to suggests a new modified materials for solidification of copper in a riverbank soil. This paper falls within the subject of this journal and in spite of the interest of present study, the manuscript needs to be improved. I doubt if a PVC column with a diameter of 1.6 cm and a length of 20 cm in the experiment can be used to simulate actual soil conditions. Is it a reference to the existing standard soil column simulation method or self-designed, if it is a self-designed experimental method, please explain why you used this method. Is it reasonable?

Validity of the findings

In results and discussion section, I suspect that pH value is not a key factor affecting migration of copper in soil? At the same time, I am very confused that the adsorption capacity of the original soil to the copper is weak, is it compared with the addition of modified materials? The amended soil has a high adsorption capacity for copper, the authors should give the mechanism explanation what is the form of EDTA modification pomace? I suggest that infrared spectrum analysis should be used to explain the reaction mechanism of functional group participation before and after adsorption, as well as before and after modification of pomace.

Additional comments

The manuscript "Effect of modified pomace on copper migration via riverbank soil in southwest China" is a study about leaching behaviour and remediation of heavy metal in riverbank soil in southwest China. This paper falls within the subject of this journal and in spite of the interest of present study, the manuscript needs to be improved:
1. The authors attempted to suggests a new modified materials for solidification of copper in a riverbank soil. Nevertheless, the authors do not explain in the Introduction Section whether the pomace or modified pomace application in copper remediation has practical value or research significance?
2. Is there heavy metal pollution, especially copper pollution, in the research area? Are raw materials (pomace) fully available?
3. Is it meaningful to use ID1.6cm, and length 20 cm of soil column experiment? The authors should give an sufficient explaination why they used this soil column and experimental method.
4. What is the form of EDTA modification pomace? how EDTA modified or combined to raw materials (pomace)?
5. Inaccurate professional descriptions of discussions (Line 156). the exchangeable state or form?
6. The initial pollution copper concentration is set to 2000 mg/L, at a 20 cm length of soil column, it is difficult to make a sufficient reaction between pollutants and soil particles, thus, the further discussion is meaningless.
7. There are some language expressions that are difficult to understand, please revise them carefully in the entire manuscript.
After these improvements the paper could be reviewed again to assess if it is suitable for publication in Peer J, however, in the present conditions I recommend Major Revisions.

---

## Round 0.2 · Minor Revisions

There are two additional corrections required:

1) In four different parts of the text (including abstract) the authors mention that "total organic carbon had negative effects on the copper contents", which suggests causality. It is more appropriate to say that total organic carbon and cupper contents are negatively correlated, since this is the conclusion obtained based on correlation analysis.

2) The legend of Table 2 says "Questionnaire" when the Table shows physical-chemical properties of the samples.

---

## Round 0.3 · Minor Revisions

Please, check the comments in the file peerj-reviewing-53356-v2_editor comments.pdf for minor corrections.

Reviewer 1 ·

Basic reporting

See comments to the editor

Experimental design

See comments to the editor

Validity of the findings

See comments to the editor

Additional comments

See comments to the editor.

Reviewer 2 ·

Basic reporting

no comment

Experimental design

After the author's revision, the methods are described with sufficient information.

Validity of the findings

After the author's revision and explanation, the result of this paper is reliable and the conclusion is appropriately stated.

Additional comments

After the author's revision and and explanation, the manuscript of article meets the Peer J. criteria and can be accepted for publication in this journal.

---

## Round 0.4 · accepted · Accept

The previous Academic Editor is no longer available and so I have taken over the handling of this submission.

Thank you for carefully addressing the comments and revising your manuscript accordingly. The paper can now be accepted for publication.